# A Cas-embedding strategy for minimizing off-target effects of DNA base editors

Yajing Liu[1,2,5], Changyang Zhou[2,5], Shisheng Huang[1,3,5], Lu Dang[4,5], Yu Wei[2,3,5], Jun He[1,3], Yingsi Zhou[2], Shaoshuai Mao[2], Wanyu Tao[1,3], Yu Zhang[1], Hui Yang [2✉], Xingxu Huang [1✉] & Tian Chi [1✉]

DNA base editors, typically comprising editing enzymes fused to the N-terminus of nCas9, display off-target effects on DNA and/or RNA, which have remained an obstacle to their clinical applications. Off-target edits are typically countered via rationally designed point mutations, but the approach is tedious and not always effective. Here, we report that the off-target effects of both A > G and C > T editors can be dramatically reduced without compromising the on-target editing simply by inserting the editing enzymes into the middle of nCas9 at tolerant sites identified using a transposon-based genetic screen. Furthermore, employing this Cas-embedding strategy, we have created a highly specific editor capable of efficient C > T editing at methylated and GC-rich sequences.

[1] School of Life Science and Technology, ShanghaiTech University, Shanghai, China. [2] Institute of Neuroscience, State Key Laboratory of Neuroscience, Key Laboratory of Primate Neurobiology, CAS Center for Excellence in Brain Science and Intelligence Technology, Shanghai Research Center for Brain Science and Brain-Inspired Intelligence, Shanghai Institutes for Biological Sciences, Chinese Academy of Sciences, Shanghai, China. [3] University of Chinese Academy of Sciences, Beijing, China. [4] Cancer Hospital and Institute of Guangzhou Medical University, Guangzhou, China. [5]These authors contributed equally: Yajing Liu, Changyang Zhou, Shisheng Huang, Lu Dang, Yu Wei. ✉email: huiyang@ion.ac.cn; huangxx@shanghaitech.edu.cn; chitian@shanghaitech.edu.cn

DNA base editors achieve targeted nucleotide substitutions without introducing double-strand DNA breaks, thus holding great potential for correcting point mutations underlying many human genetic diseases[1,2]. However, both adenine base editor (ABE) and cytosine base editor (CBE) can create substantial off-target edits on RNA[3,4], whereas CBE additionally has off-target effects on DNA[1,3–8]. Such effects raise safety concerns regarding the clinical use of the base editors, which have motivated intense searches for countermeasures[1,9]. Off-target editing results from the intrinsic properties of the editing enzymes harnessed for base editing, and consequently, have been countered by mutating these enzymes. Multiple high-specificity editors bearing rationally engineered point mutations as well as deletions at the deaminase moiety have been reported, including YE1 and ABE^F148A [3,4,10–14]. However, this conventional strategy for minimizing the off-target effects requires prior knowledge about the enzyme structure, is labor intensive, and not always effective. Alternative approaches that are straightforward and generally applicable would be highly desirable.

It has been observed that some editing enzymes (APOBEC1 and Tad-TadA*) produce more off-target edits on RNA in HEK293T cells when expressed as free proteins than as N-terminal fusion protein to nCas9[4], suggesting that fusing the enzymes to nCas9 may hinder off-target editing perhaps due to steric hindrance. We hypothesized that inserting the enzymes into the middle of nCas9, rather than fusing it to its N terminus, might further reduce the off-target effects, thus offering an attractive alternative to the mutagenesis approach. To test this idea, we first designed a genetic screen to systematically identify the nCas9 sites tolerant of adenine deaminase insertion, and subsequently demonstrated that the off-target editing of both ABE and CBE can be dramatically repressed simply by relocating the deaminase moiety in the base editors from the N terminus of nCas9 to the tolerant sites in the middle of nCas9, as described below.

## Results

### Genetic screen for Cas9 sites tolerant of TadA-TadA* insertion.

We sought to use MuA-transposon-based genetic screen[15,16] to identify the nCas9 sites tolerant of deaminase insertions. First, we constructed an "all-in-one" plasmid expressing: (1) nCas9 under the control of isopropyl β-D-1-thiogalactopyranoside (IPTG)-inducible promoter, (2) the ampicillin-resistant (AmpR) gene bearing a C > T mutation that created a premature stop codon (A118X), and (3) a single-guide RNA (sgRNA) under the control of the J23119 promoter for repairing the premature stop codon. The bacteria transformed with this plasmid would be ampicillin sensitive until A > G editing occurs on the bottom strand of DNA, which corrects the C > T mutation in the top strand to restore the translation (Fig. 1a and Supplementary Fig. 1a). We next used Mu transposon to randomly insert the DNA encoding TadA-TadA* into the "all-in-one" plasmid. This insertion plasmid library, prepared in vitro, was then electroporated into *Escherichia coli* and the cells were grown overnight on plates containing kanamycin but no ampicillin. We evaluated the insertion efficiencies at various positions on nCas9 by deep-sequencing the unscreened plasmid library extracted from the recovered cells. At the nCas9 coding sequence, we found 51,393 insertions, with at least one insertion in 99% of amino acid positions (Fig. 1b and Supplementary Table 2), demonstrating that the Mu-mediated mutagenesis was efficient and unbiased. As expected, insertions were hardly detectable in the Kan resistance gene and the replication-related f1 region (Fig. 1b). IPTG was added to the mixture to induce the fusion protein expression (nCas9-TadA-TadA*) before the cells were transferred to plates

with ampicillin to select the cells expressing the repaired AmpR gene. Positive clones were picked, and the plasmids were extracted and sequenced to determine the editing efficiencies and TadA-TadA* insertion sites (Fig. 1c).

In total, 43 insertional sites were found on nCas9 by analyzing the plasmids extracted from the recovered ampicillin-resistant colonies. Most of the central fusion ABE variants achieve robust A-to-G editing at the premature stop codon (Fig. 1c). The ABE variants with TadA-TadA* inserted into these highly tolerant sites are identical to ABEmax[17], except for the location of TadA-TadA*, and were termed CE-ABEs ("CE" stands for Cas embedding). Among the 43 insertion sites recovered in the screened library, nine sites were clustered together within a short (16-amino acid (a.a.)) segment, occurring at 1048Thr, 1050Ile, 1051Thr, 1052Leu, 1054Asn, 1056Glu, 1057Ile, 1059Lys, and 1063Ile. The enrichment of these sites in the screened library was specific, because in the unselected library, these sites were inserted only 61, 39, 90, 38, 5, 29, 76, 53, and 25 times, respectively, much less frequently than some other sites (e.g., 1090Pro, inserted 280 times, Supplementary Table 2) that were not recovered after screening. Thus, the 16-a.a. fragment was highly tolerant of insertion and presumably dispensable for nCas9 function. Indeed, this fragment, harbored inside the RuvC III domain in the NUC lobe[18], is not conserved among 28 SpCas9 orthologs (Supplementary Fig. 1b). Therefore, we replaced the 1048Thr-1063Ile region with TadA-TadA* to generate CE-ABE^1048−1063.

### Performance of CE-ABEs in HEK293T cells.

We next tested the 20 most frequently recovered CE-ABEs in HEK293T cells. At an endogenous site containing multiple As within the editing window (Site1), 12 of the 20 CE-ABEs were as active as ABEmax, with editing efficiencies ranging from 66–89% as compared with 86% for ABEmax (Fig. 2a). The editing efficiencies of the various CE-ABEs in HEK293T cells were largely consistent with their recovery rates in prokaryotic cells, demonstrating the robustness of the screen. We then examined the off-target RNA editing for the 12 variants at three RNA off-target sites known to be highly susceptible to ABEmax. All 12 CE-ABEs showed remarkable reductions in the editing on at least two of the three sites, and four of the 12 variants at all three sites (Fig. 2b, the four variants marked in red). We further used RNA-sequencing (RNA-seq) to profile the off-target edits by these top 4 CE-ABEs at the entire transcriptome, and compared them with ABEmax and ABE7.10^F148A. ABEmax induced massive off-target edits as reported before, which was reduced at least 6× in CE-ABEs, with CE^1072-ABE as much as 236× (single-nucleotide variant (SNVs) reduced from 20,739 to 88, a level similar to that produced by ABE7.10^F148A ; Fig. 2c). In contrast, on-target editing by ABEmax was either comparable (92% and 89%, for ABEmax and CE^1048−1063-ABE, respectively) or only mildly reduced (to 75% and 73% for CE^776 -ABE or CE^1263-ABE, respectively), except that for CE^1072-ABE, the activity was reduced markedly (to 33%) (Fig. 2d).

The data above established CE^1048−1063-ABE as the optimal CE-ABE with balanced efficiency and specificity. Therefore, we characterized its performance further, at 8 and 12 randomly selected genomic sites in HEK293T cells (Fig. 2e, f) and mouse N2a cells (Fig. 2g, h), respectively. We found that CE^1048−1063-ABE was comparable to ABEmax in terms of editing rates (Fig. 2e, g, summarized in Fig. 2i), while the editing window was slightly enlarged (Fig. 2f, h). We concluded that embedding an adenine deaminase in nCas9 could markedly reduce the off-target effects on RNA with only a minimal impact on on-target editing.

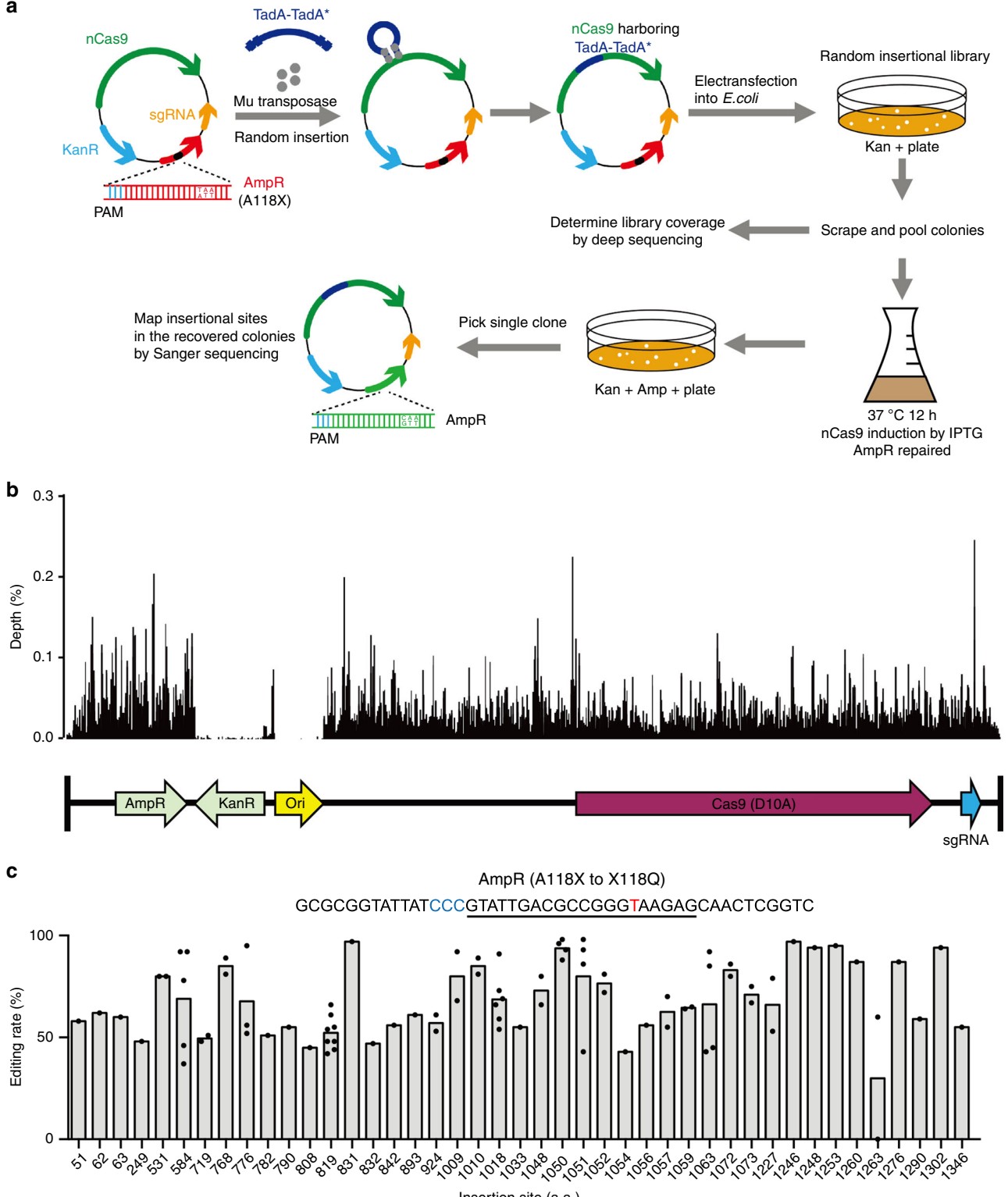

**Fig. 1 Genetic screens for tolerant sites on nCas9 using transposon-mediated random mutagenesis. a** Workflow. AmpR (A118X), ampicillin resistance gene bearing a premature stop codon. **b** Insertion site distribution pattern across the entire plasmid in the unscreened library, revealed by deep-seq. **c** nCas9 insertion sites in the screened library and the corresponding A > G conversion rates at the Amp resistance gene in the same plasmid, as revealed by Sanger sequencing. Each dot represents a single Amp-resistant colony, with a total of 84 colonies examined from three independent experiments. The AmpR sequence surrounding A118X is shown, with the targeted base and PAM highlighted in red and blue, respectively. A:T to G:C editing at the target would convert the stop codon to CAA (encoding Q).

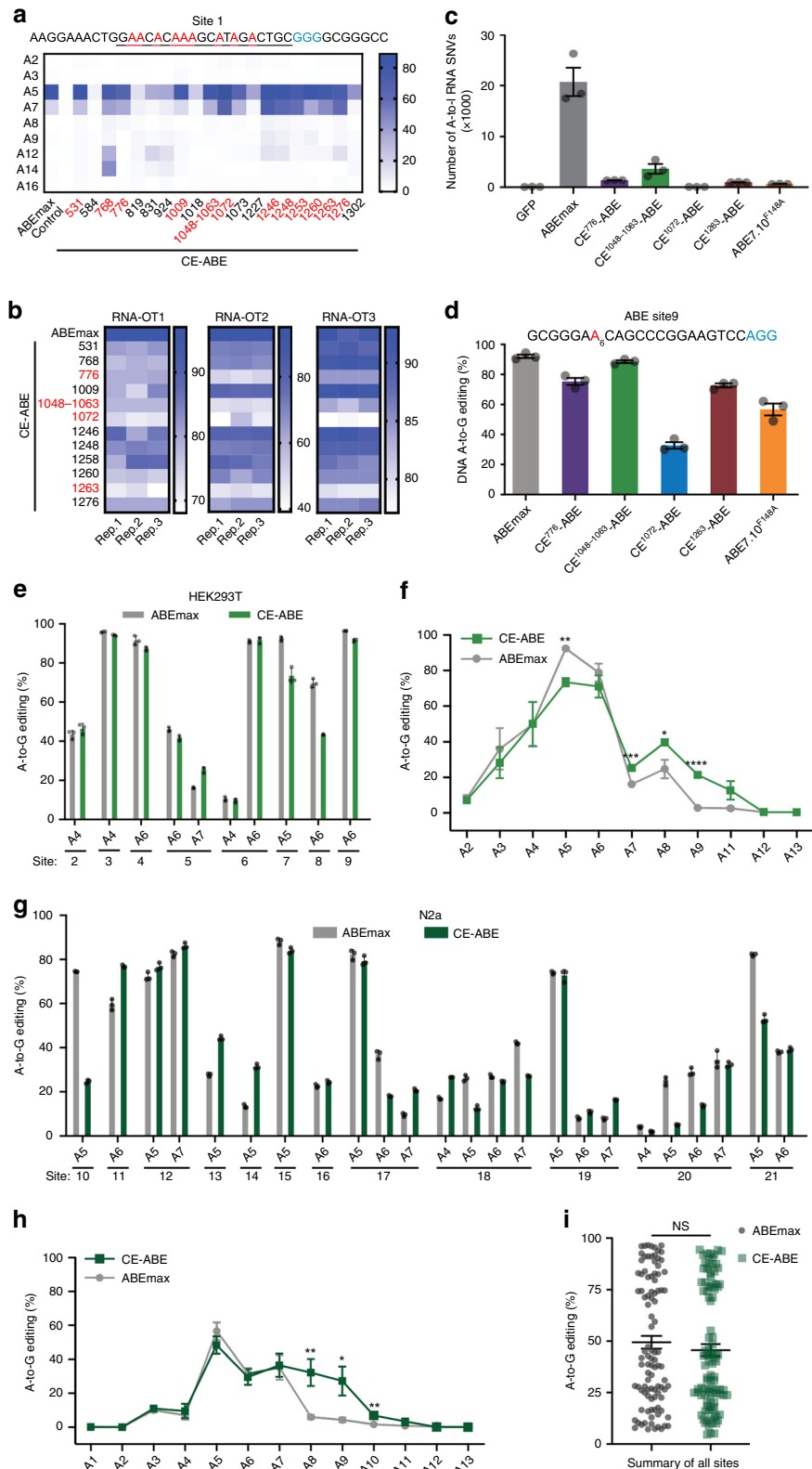

**The CE strategy is applicable to CBEs bearing different Apobec proteins.** Encouraged by the success in CE-ABEs, we sought to derive the CE versions of two distinct CBEs, the first being AncBE4max consisting of the APOBEC ancestor Anc689 linked to the N terminus of nCas9, which is highly active but presumably also highly non-specific[17]. To create the CE version of AncBE4max, we relocated Anc689 to position 1048–1063 in nCas9,

replacing the native Cas9 sequence in the process; the resulting editor was termed CE[1048–1063]-CBE. We also generated CE[1071]-CBE. The second CBE we sought to optimize involves BE4-A3A, comprising APOBEC3A fused to the N terminus of nCas9. BE4-A3A is one of the most active CBEs created so far, but also displays the highest guide RNA (gRNA)-independent off-target editing on DNA[11,19]. We have previously found that introducing

**Fig. 2 Characterization of CE-ABEs in mammalian (HEK293 and N2a) cells. a** On-target editing in HEK293T cells by 20 CE-ABE variants at Site1, a known ABE target[24]. Twelve CE-ABEs were as active as ABEmax (red). Data averaged from three biological replicates. **b** Off-target editing by the 12 CE-ABEs at three known RNA off-target sites of ABEmax we identified using RNA-seq in preliminary studies. Top four candidates with minimal off-target effects are highlighted (red). **c, d** On-target (**c**) and off-target (**d**) editing by the four candidates. **e, f** On-target editing by the optimal variant (CE$^{1048-1063}$-ABE) benchmarked against ABEmax in HEK293T cells. Shown are the editing rates of the susceptible As at individual sites (**e**) and at all the As along the target site specified by the gRNA protospacers, as averaged from all the eight sites (**f**). $P = 0.0014$, 0.0005, 0.0199, and 0.0001 for A5, A7, A8, and A9, respectively. **g, h** Same as panels (**e**) and (**f**), except that the editors were compared in N2a cells. $P = 0.0048$, 0.0127 and 0.0021 for A8, A9, and A10, respectively. **i** Editing efficiencies on 20 sites, summarized from panels (**e**) and (**g**). NS not significant ($P = 0.36$). In **a–h**, the 20 test sites used (#2–21) are described in Supplementary Table 1, and the values in the graphs are mean ± s.e.m. ($n = 3$ independent biological replicates), with *, **, *** and **** representing $P < 0.05$, 0.01, 0.001 and <0.0001, respectively (two-sided unpaired $t$ test).

Y130F into A3A in the context of BE3-A3A mitigates the off-target effect only partially, with substantial off-target edits persisting in BE3-A3A(Y130F), which limits the use of the editor[10]. To minimize the off-target effect of A3A(Y130F), we replaced rA1 of CE$^{1048-1063}$-CBE with A3A(Y130F), generating CE$^{1048-1063}$-A3A(Y130F). We also generated CE$^{1072}$-A3A(Y130F) and BE4max-A3A(Y130F) for comparison.

We first determined the effects of CE on on-target editing. To this end, we compared the on-target editing rates of CE$^{1048-1063}$-CBE with that of AncBE4max at nine randomly selected target sites in HEK293T cells, finding that CE did not compromise the editing efficiency of AncBE4max (Fig. 3a, b) or alter its editing window (Fig. 3c). A similar result was seen when comparing CE$^{1048-1063}$-A3A(Y130F) with BE4max-A3A(Y130F) (Fig. 3d–f). A unique advantage of BE-A3A is that they can efficiently edit GC-rich and highly methylated regions, contrary to the traditional CBEs utilizing rA1[19]. To determine whether this important advantage of A3A is retained in CE$^{1048-1063}$-A3A (Y130), we benchmarked CE$^{1048-1063}$-A3A(Y130) against YE1-BE4max, the most active CBE we and others reported[10,12]. We compared the two editors in HEK293T cells at three highly methylated target genes. Indeed, CE$^{1048-1063}$-A3A(Y130F) clearly outperformed YE1-BE4max at all three sites (Fig. 3g, h), confirming that CE$^{1048-1063}$-A3A(Y130F) was preferable over YE1-BE4max for highly efficient editing at methylated (and by inference, GC-rich) regions. Of note, at these sites, CE$^{1048-1063}$-A3A(Y130F) was as active as BE4max-A3A(Y130F), but more active than BE3-A3A(Y130F), as expected (Fig. 3g, h).

Having demonstrated the CE did not alter the on-target editing by CBEs, we next determined whether CE could indeed selectively inhibit their off-target effects. In contrast to ABEs, which do not affect the genome, CBEs are known to display gRNA-independent off-target editing on the genome in addition to off-target effects on the transcriptome. We first evaluated the DNA off-target edits using GOTI (genome-wide off-target analysis by two-cell embryo injection), a highly sensitive and physiologically relevant assay for detecting random genomic off-target edits. In this method, the editors were coexpressed with a sgRNA targeting the *Tyrosinase* gene, and the DNA from E12.5 embryos was sequenced at a depth of ~30×. AncBE4max created ~773 SNVs per embryo, 43× above the background in the GFP control (18 SNVs) (Fig. 4a, top; Supplementary Fig. 2a, b). In sharp contrast, only 19 and 31 SNVs per embryo were detected in the embryos treated with CE-CBE$^{1048-1063}$ and the CE-CBE$^{1072}$, respectively, each similar to the GFP control and to YE1-BE3, a highly specific BE3 variant[10,12] (Fig. 4a, top; Supplementary Fig. 2a, b). As expected, in the SNVs detected in AncBE4max-treated embryos, the SNV subsets resulting from C:G > T:A conversion predominated, constituting 92% of the total SNVs, whereas in the embryos treated with CE-CBEs, GFP, or YE1-BE3, the subsets were less abundant (<68%, but accurate quantification not feasible due to the scarcity of SNVs; Fig. 4a, bottom and Supplementary Fig. 2c). The same trend was observed for A3A-

editors, with substantial numbers of SNVs (276) detected in BE3-A3A(Y130F) but much less (43–63) in CE$^{1048-1063}$-A3A(Y130F) or CE$^{1072}$-A3A(Y130F) (Fig. 4a and Supplementary Fig. 2). We conclude that CE markedly decreased DNA off-target effects of CBEs. Importantly, this decrease is not an artifact resulting from nonspecific inactivation of the CE-editors, as revealed by their robust editing at the on-target in the mouse embryos (Fig. 4b).

We next quantified off-target RNA editing using RNA-seq. BE4max created massive off-target edits (~3000) as expected, but interestingly, AncBE4max only 74 edits, presumably reflecting the property of Anc689 harnessed in the editor (Fig. 4c). Nevertheless, the off-target effect was clearly above the GFP control, and was completely eliminated in CE-CBE, which produced only 15 edits, a level indistinguishable from the background (Fig. 4c). For A3A editors, CE-mediated suppression of RNA off-target editing was much more pronounced, reducing the SNV numbers 100×, from 2025 in BE4max-A3A(Y130F) to only 18 in CE$^{1048-1063}$-A3A(Y130F), a level only slightly above the GFP background (11) (Fig. 4d).

These results show that the CE strategy is applicable to CBEs, enabling the creation of CE$^{1048-1063}$-A3A(Y130F), a highly specific editor capable of robust editing at methylated or presumably GC-rich sequences.

## Discussion

We have shown that CE is a powerful countermeasure of off-target effects of DNA base editors, which is much simpler than the conventional method of deaminase domain mutagenesis[3,4,10–14]. Although the off-target effects can also be countered by mutations, these mutations are identified on a case-by-case basis, typically via extensive structure–function analysis, contrary to CE, which requires no prior knowledge about the enzymes. We are aware that CE may be insufficient for fully eliminating off-target effects for some editors, but so may be mutations, and in this case, the two approaches may be used in conjunction to achieve the optimal effect, as illustrated in CE-A3A(Y130F), where CE and Y130F act together to minimize the off-targets on DNA. Indeed, CE is not mutually exclusive but complementary with other editor optimization methods, including the use of short-rigid linkers, an elegant strategy for narrowing the editing window[13,14]. Of note, we have recently found that for an RNA base editor comprising the ADAR2 deaminase domain fused to the programmable RNA-binding protein dCasRx, inserting the deaminase domain into a flexible loop on the surface of dCasRx could reduce the off-target effects on RNA slightly (1.8×) relative to a terminal fusion configuration[20]. The data suggest that the CE method is perhaps generalizable to RNA base editors, assuming that the transposon-based screen on dCasRx can similarly yield optimal insertion sites analogous to the ones discovered in nCas9.

How exactly CE reduces the off-target effects is unclear. Off-target editing of DNA base editors is caused by the deaminase moiety independent of nCas9, whereas the on-target editing is achieved when the deaminase moiety is brought into the proximity

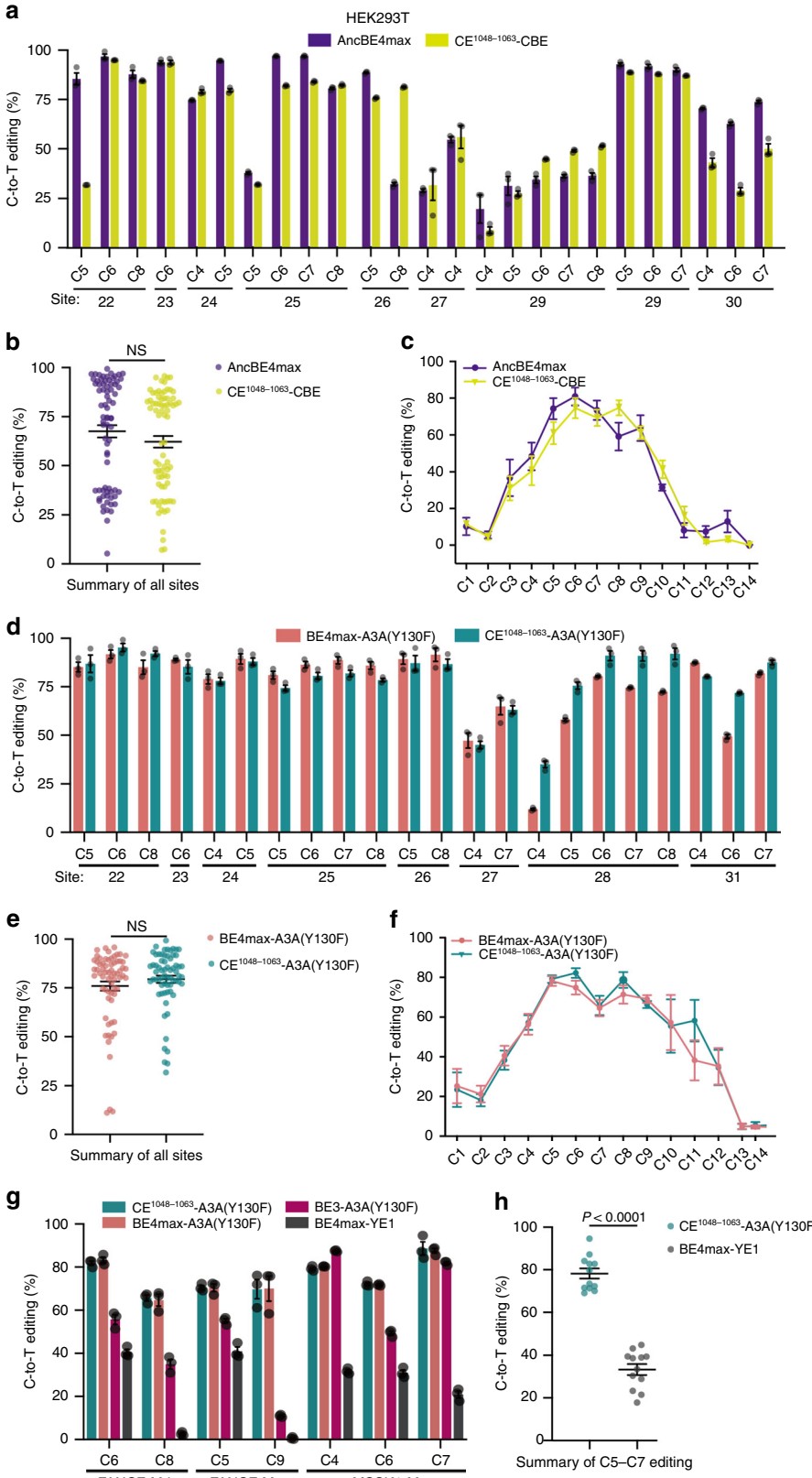

**Fig. 3 On-target editing by CE$^{1048-1063}$-CBE and CE$^{1048-1063}$-A3A(Y130F) in HEK293T cells. a–f** CE$^{1048-1063}$-CBE was benchmarked against AncBE4max (**a–c**), while CE$^{1048-1063}$-A3A(Y130F) against BE4-A3A(Y130F) (**d–f**). The editing rates of the susceptible Cs are shown for individual sites (**a**, **d**) and as a pool (**b**, **e**). Also shown are the editing rates of all the Cs along the target region specified by the gRNA protospacers, as averaged from all the tested sites (**c**, **f**). The test sites used (#22–31) are described in Supplementary Table 1. **g**, **h** CE$^{1048-1063}$-A3A editing at representative methylated sites[14]. The editing rates at susceptible Cs are displayed for individual sites (**g**) or as a pool (**h**). Values in the **a**, **c**, **d**, **f**, **g** are mean ± s.e.m. from three biological replicates. NS not significant.

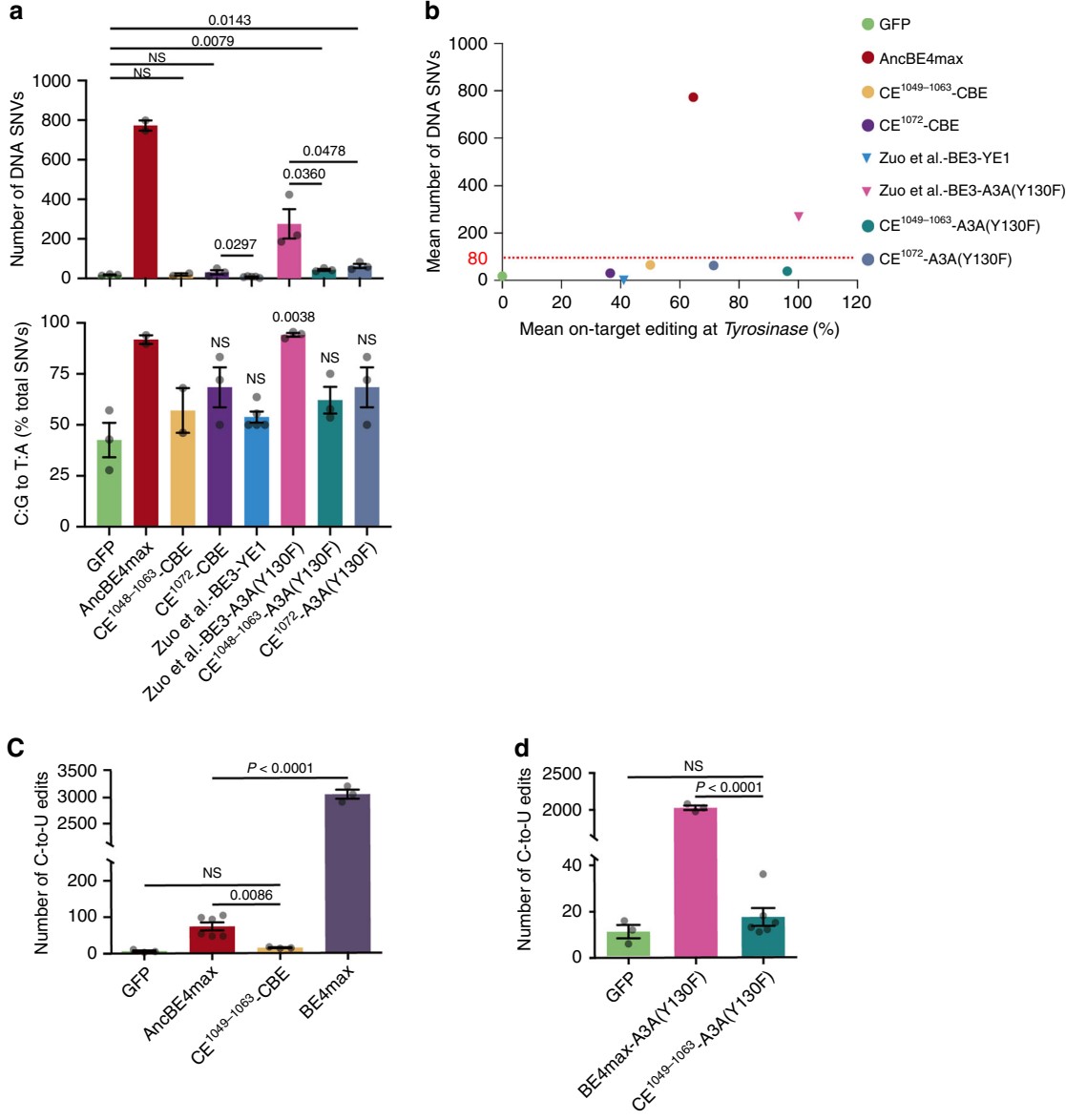

**Fig. 4 Off-target effects of CE-CBEs. a**, **b**, GOTI assay detecting off-target SNVs at the mouse genome. The bar graphs (**a**) display the total numbers of SNVs (top) and the proportions of the relevant subsets (bottom), while the scatter plot (**b**) displays the total SNV numbers and the on-target editing rates of the editors. The data were all generated in this study except for YE1-BE3 and BE3-A3A(Y130F), which were from refs. [5],[10], respectively. Values in **a**, **b** are mean ± s.e.m from two to three biological replicates as indicated by the dots. **c**, **d** RNA-seq analysis of off-target effects at the transcriptome for CE-CBE (**c**) and CE-A3A (**d**). Values in **c**, **d** are mean ± s.e.m for three or six biological replicates as indicated by the dots. NS not significant.

of the targets by nCas9. Embedding the deaminase domains in nCas9 might create some steric hindrance that hampers their free access to the off-targets, but not an on-target owing to their forced tethering via nCas9. This idea is testable by replacing the current XTEN linkers in the CE editors with longer, flexible linkers, which could relieve the putative steric hindrance. Besides, structural studies as exemplified by Doudna and colleagues[21] should be valuable for providing conclusive evidence regarding the mechanisms of CE. Of note, circularly permutated nCas9 has been created whose protein termini are placed at various positions[22]. It would be interesting to determine whether fusing the deaminase domains to the termini could also selectively compromise off-target editing.

We have used CE to successfully improve the specificity for both ABE and CEB. In particular, CE[1048–1063]-ABE is as efficient as ABE7.10[F148A] but slightly less specific, and so unable to outperform ABE7.10[F148A]. However, CE-ABE is far more specific than ABE-max, and our strategy for generating CE-ABE is very simple,

contrary to the mutagenesis approach used to engineer ABE7.10[F148A]. In contrast to CE-ABE, CE[1048–1063]-A3A(Y130F) has unique advantages over the existing CBEs, being highly efficient at GC-rich and methylated sites while displaying little off-target editing on genomic DNA. As described above, this novel editor illustrates how CE and other strategies can be used together for base editor optimization.

Finally, we have designed a transposase-based platform and successfully used it to identify 43 sites in nCas9 tolerant of deaminase insertion. This information may be useful for optimizing other base editors such as the prime editor[23]. Interestingly, in a previous study, Doudna and colleagues delineated 175 hotspots on dCas9 tolerant of insertions of a PDZ domain[16], but there is little overlap between these hot spots and the 43 CE sites mapped in this study. This discrepancy might reflect a "sampling error" due to unsaturated screening. However, this scenario seems unlikely. We have performed a total of three screens, obtaining

45, 40, and 37 colonies, respectively. Importantly, some insertion sites were recovered multiple times in almost every screen, suggesting that the screening has been saturating. We therefore favor the possibility that the discrepancy instead reflects the differences in the experimental design between the two studies, as Doudna and colleagues searched for dCas9 sites tolerant of PDZ insertion, and their screen readout is dCas9-mediated gene silencing. The two studies collectively suggest that although Cas9 tolerates insertions at numerous sites, the optimal sites may be protein-specific. Nevertheless, the region in Cas9 between a.a. 1048 and 1063 identified in our study may be generally tolerant of diverse proteins, because this site is non-conserved and supportive of superior performance for both ABE and CBE. This region could therefore serve as the starting point when applying the CE strategy to new proteins in the future.

## Methods

**Plasmid construction**. Primers and plasmids are listed in Supplementary Table 1. pCMV-nCas9-KanR-AmpR(A118X)-sgRNA, the all-in-one plasmid for insertion screening, was assembled from pCMV-ABEmax (Addgene 112095), pUC57-Kan (Addgene, 51132), and pGL3-U6-sgRNA (Addgene, 51133). The sgRNA expression vector for mammalian cells was constructed using BsaI-digested pGL3-U6-sgRNA-EGFP with annealed DNA oligos (Supplementary Table 1). The sgRNA expression vector for GOTI was constructed by cloning annealed DNA oligos (Supplementary Table 1) into BbsI-digested pUC57-sgRNA (Addgene, 51132). CE-ABEs and CE-CBEs were derived from pCMV-ABEmax (Addgene 112095) and pCMV-AncBE4max (112094), respectively.

**Insertional library construction, characterization, and screening**. TadA-TadA* was PCR-amplified from pCMV-ABEmax (Addgene 112095) and cloned into a MuA-transposon vector (Supplementary Fig. 1a). The transposon was excised from the vector using BsaI digestion before random insertion into pCMV-nCas9-KanR-AmpR(A118X)-sgRNA in an in vitro reaction containing 250 ng transposon, 500 ng nCas9 plasmid, and 1 μL of MuA transposase (F-701, Thermo Fisher). The reaction was incubated at 30 °C for 1 h to achieve random insertion, followed by 75 °C for 10 min to inactivate the MuA transposase. The DNA was precipitated, resuspended in 5 μL deionized water, and electroporated into 100 μL BL21(DE3)-electrocompetent cells (Shanghai Weidi Biotechnology, EE1002). A total of 1 mL SOC media were then added and the bacteria were cultured at 37 °C for 1 h. The cells were then plated out on several LB agar plates containing 10 μg/mL kanamycin and incubated at 37 °C overnight. The colonies were then collected by scraping, characterized, and screened. Specifically, to characterize the unscreened library, the insertional sites were deep-sequenced using an Illumina HiSeq X Ten (2 × 150 PE) at the Novogene Bioinformatics Institute (Beijing, China). All cleaned reads were first mapped to the backbone sequence using BWA v0.7.16 with the default parameters. The soft clipped reads were extracted and then mapped to the insertion sequence. All mapped soft clipped reads were checked and the breakpoints were recorded as insertion sites. To screen the library, the scraped cells were resuspended in 100 mL LB containing 500 μM IPTG. The culture was incubated for 12 h to induce nCas9 expression and repair the AmpR (A118X) mutation. Decreasing amounts of the culture (5 mL, 1 mL, 500 μL, 100 μL) were then plated out on 15-cm LB agar plates supplemented with ampicillin (10 μg/mL) and kanamycin (10 μg/mL). After an overnight incubation, the colonies were picked and subjected to Sanger sequencing to evaluate base editing at AmpR (A118X) and to determine the TadA-TadA* insertion sites.

**Cell culture and transfection**. HEK293T (ATCC CRL-3216) and Neuro-2a (N2a) (ATCC HTB-96) cells were cultured in Dulbecco's modified Eagle's medium (DMEM) (Hyclone, SH30243.01) supplemented with 10% fetal bovine serum (v/v) (Gemini, 900-108) and penicillin and streptomycin (Gibco, 15140122). Cells were passaged once every 3 days and incubated at 37 °C with 5% $CO_2$. All cells used in the experiment have been tested to exclude mycoplasma contamination. To evaluate the CE-ABEs, HEK293T cells were seeded on poly-D-lysine- (Sigma, P4707) coated 12-well plates (JET-BIOFIL, TCP010012) and transfected about 14 h at ~80% density as per the manufacturer's protocols (Thermo Fisher Scientific, 11668019). Editor-expressing vectors (700 ng) were co-transfected with corresponding sgRNA-GFP plasmids (300 ng), and the cells with the highest 5% of GFP signal isolated by fluorescence-activated cell sorting 48 h later. However, for the experiment in Fig. 2a, b, "all-in-one" plasmids (1 μg) expressing CE-ABE-P2A-GFP together with gRNA targeting Site1 was transfected instead. The isolated cells were analyzed for DNA and RNA editing as described below.

**On-target genome editing in HEK293T and N2a cells**. Genomic DNA was extracted using QuickExtract™ DNA Extraction Solution (Lucigen). The

fragments encompassing the target sites (~200 bp) were PCR-amplified using Phanta Max Super-Fidelity DNA polymerase (Vazyme, P505-03); the primers used are listed in Supplementary Table 1. The amplicons were analyzed by deep sequencing on the Illumina Hiseq X Ten (2 × 150 PE) platform. The adapter pair of the paired-end reads were removed using AdapterRemoval version 2.2.2, and paired-end read alignments of 11 bp or more bases were combined into a single consensus read. All the processed reads were then mapped to the target sequences using the BWA-MEM algorithm (BWA v0.7.16). For each site, the mutation rate was calculated using bam-read count with parameters $-q$ 20 $-b$ 30. Indels were calculated based on the reads containing at least one inserted or deleted nucleotide in the protospacer. Indel frequency was expressed as the number of indel-containing reads/total mapped reads.

**RNA off-target editing in HEK293T cells**. Cells were lysed in TRIzol reagent (Invitrogen, 15596026) and total RNA was extracted. For evaluation of off-target editing at specific sites, mRNA was reverse transcribed using HiScript II Q RT SuperMix Kit (Vazyme, R223-01) and the predicted off-targets were amplified using Phanta Max Super-Fidelity DNA polymerase (Vazyme, P505-03) with the primers listed in Supplementary Table 1. The amplicons were analyzed by Sanger sequencing and the editing rates were calculated using the following tool: https://moriaritylab. shinyapps.io/editr_v10/. For profiling of global RNA off-target editing, RNA samples were sequenced using an Illumina HiSeq X Ten (2 × 150 PE) at the Novogene Bioinformatics Institute (Beijing, China), at a depth of ~20 million reads per sample. The reads were mapped to the human reference genome (hg38) by STAR software (Version 2.5.1); annotation from GENCODE version v30 was used. After removing duplication, variants were identified using GATK HaplotypeCaller (version 4.1.2) and filtered with quality by depth) < 2. All variants were verified and quantified by bam-readcount with parameters $-q$ 20 $-b$ 30. The depth for a given edit had to be at least 10× and these edits were required to have at least 99% of reads supporting the reference allele in the wild-type samples. Finally, only A-to-G (for ABEs) or C-to-T (for CBEs) edits in the transcribed strand were considered for downstream analysis.

**GOTI assay**. gRNA targeting the *Tyrosinase* gene and the mRNA encoding the editors were prepared as described[4]. Briefly, the gRNA was transcribed in vitro using the TranscriptAid T7 High Yield Transcription Kit (Thermo Fisher Scientific) with the primer listed in the Supplementary Table 1. sgRNAs were purified using the MEGAclear Kit (Thermo Fisher Scientific). The editor mRNAs were transcribed using mMESSAGE mMACHINE T7 Ultra Kit (Ambion, Life Technologies, AM1345) and purified using RNeasy Protect Mini Kit (Qiagen, 74124). All the subsequent steps were also performed as previously described[4], but with one important modification in that the edited and non-edited blastomeres are now independently grown in distinct foster mothers to help ensure the healthy development of the edited embryos. Specifically, we obtained two-cell embryos on the B6D2F1(C57BL/6 × DBA2J) background, transplanted each blastomere to a recipient zona pellucida, injected the RNA mixture into one of the blastomeres, and transplant both the injected and control blastomeres into a respective foster ICR (albino) mother together with 6–8 ICR (albino) embryo. The embryos were analyzed at E12.5 by targeted sequencing for on-target editing and by whole-genome sequencing (WGS) for gRNA-independent off-target editing as usual. Briefly, WGS was performed at mean coverage of 30× using an Illumina HiSeq X Ten. Raw reads were trimmed with Trimmomatic (v0.36) and duplicates were removed using Sambamba (v0.6.7) before mapping qualified reads to the mouse reference genome (mm10) using BWA (v0.7.16). Three algorithms, Mutect2 (v3.5), Lofreq (v2.1.2), and Strelka (v2.7.1), were used to identify de novo variants, with the paired non-injected sample in the same embryo serving as a control. The subset of the SNVs reported by all three algorithms was considered the true variants. The gRNA-dependent potential off-target sites bearing the NRG PAM were predicted using CasOT (http://casot.cbi.pku.edu.cn/) and Cas-OFFinder (http://www.rgenome.net/cas-offinder/). The filter setting for CasOT was ≤2 and ≤3 mismatches for the seed and non-seed regions, respectively, whereas the setting for Cas-OFFinder was ≤3 mismatches for the 20-nucleotide protospacer.

**Statistical analysis**. No statistical methods were used to predetermine sample sizes. All values are shown as mean ± s.e.m. Unpaired Student's *t* test (two-tailed) was used for comparisons and $p < 0.05$ was considered to be statistically significant.

**Reporting summary**. Further information on research design is available in the Nature Research Reporting Summary linked to this article.

## Data availability

High-throughput sequencing data are available in the NCBI Sequence Read Archive database (https://www.ncbi.nlm.nih.gov/bioproject/?term=PRJNA660112). Source data are provided with this paper.

 ARTICLE

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

## Acknowledgements
We thank Haibo Zhou for discussion and Yidi Sun for help with bioinformatics. We also thank the Molecular and Cell Biology Core Facility (MCBCF) at the School of Life Science and Technology, ShanghaiTech University for providing technical support. This work is supported by National Key R&D Program (2016YFA0500903) and National Science Foundation of China (81830004).

## Author contributions
Y.L., C.Z., and S.H. conceived of the project under the guidance of H.Y., X.H., and T.C. Y.L., C.Z., and S.H. performed the genetic screen. Y.L., L.D., and W.T. characterized the editors in vitro with help from S.M. and Y.Z., Y.W., C.Z., and Y.L. performed the GOTI experiments. S.H. and J.H. performed bioinformatics analysis with help from Y.S.Z., Y.L., and T.C. analyzed the data and wrote the manuscript with input from S.H. and C.Z.

## Competing interests
The authors declare no competing interests.
