## [Peer Review File · Nature Communications]

Reviewers' Comments:

Reviewer #1:

Remarks to the Author:

The authors of this manuscript have tested the interesting idea that Cas9-internal insertion of deaminase domains (rather than N- or C-terminal fusions) can produce base editors with superior properties. Neutral insertion sites were identified by an elegant transposon-based genetic screen. Testing both cytidine and adenosine deaminases, the authors were indeed able to create base editors with reduced off-target effects with respect to both RNA editing and DNA editing. Overall, I find the approach developed by the authors very interesting. The new base editors certainly add useful tools to the genome editing toolbox, and, in the future, the approach likely can be extended to other Cas fusion proteins.

Major points:

1. The authors tested their approach as a strategy to potentially reduce the off-target effects of BEs. However, all generated CE-ABEs do not outperform the previously reported ABE7.10F148A in terms of reducing RNA off-targets or improving on-target editing efficiency. Considering that ABE7.10F148A and the new CE-ABEs also have different base editing windows, it seems desirable to test a few more target sites and compare editing efficiency and precision.
2. Figure 1a: How many colonies were obtained by the double selection and how many of them were sequenced? This is important to know to be able to judge whether or not the screen has been saturating. Also, Oakes et al. (2016) employed a similar screening strategy and also identified tolerant insertion sites. The combined information on tolerant sites should be discussed with regard to possible saturation and future engineering options.
3. In Figure 2, the authors compare the editing efficiencies of CE1048-1068-A3A(Y130F) and YE1-BE4max in a C-rich sequence. Considering that the YE1 variant has recently been reported to reduce DNA off-targets, it is unfortunate that the authors have not included YE1 as a comparator for DNA off-targets.

Minor points:

1. In Figure 1a, the authors write "LB with Kan 37 °C 5h". In the text, they write "cells grown overnight on plates" (l. 43) and in the Methods section it is 16 hours (l. 199). These inconsistencies should be removed.
2. Line 96: The GOTI method was developed by Zuo et al., (Science, 2019), which should be cited here.
3. There are many other instances, where it would be appropriate to cite additional references. For example, in the last sentence of the text, prime editors are mentioned without a reference (Anzalone et al., Nature 2019). Also, previous work on the design of base editors with improved properties should be discussed in the Introduction and/or Discussion section of the manuscript (e.g., Kim, Y. B., et al. Increasing the genome-targeting scope and precision of base editing with engineered Cas9-cytidine deaminase fusions. Nat. Biotechnol. 35, 371-376, 2017; Tan, J., et al. Engineering of high-precision base editors for site-specific single nucleotide replacement. Nat. Commun. 10, 439, 2019; Tan, J., et al. Expanding the genome-targeting scope and the site selectivity of high-precision base editors. Nat. Commun. 11, 629, 2020). Overall, the writing is quite compact, and clarity and intelligibility would benefit from a few extra sentences here and there.
4. Figure 1e: lettering is too small to be legible (especially the letters and numbers in subscript and superscript). Also, in the legend, it is unclear what the "n" refers to.
5. Fig. S1d: ditto – font size needs to be increased.
6. Fig. S6: What does Tyr mean?
7. Line 100, "Supplementary Fig 6a" should be "Supplementary Fig 6".
8. Line 122, "A3A(Y130)" should be "A3A(Y130F)".
9. Line 129, "mutagesis" should be "mutagenesis", l. 15, "an obstacle", l. 257, "zona pellucida" (not: zonae pellucidae), l. 141: "prime editor" (not: primer editor), etc. – there are numerous other small language-related problems, and overall, the manuscript could do with a final read by a native English speaker.
10. Line 130: To not confuse the reader, it would be better to refer to point mutations "in the

deaminase domain". Also, deaminase truncations should be mentioned here (Tan, J. et al., 2019 and 2020).

11. L. 98: "sgRNA targeting Tyr" – do the authors mean "a codon for Tyr"?

12. L.75f.: I don't think a 10% reduction in on-target editing can be called "uncompromised"

13. L.106: A 30-40% reduction in off-target mutations should not be referred to as "dramatic"

Reviewer #2:

Remarks to the Author:

General comments;

DNA base editors, which include cytosine base editors (CBEs) and adenine base editors (ABEs), are promising tools and have been harnessed to many research areas such as therapeutic editing. However, several studies showed shortcomings of the tools, including promiscuous deamination effects in both DNA and RNA (CBEs) and in DNA (ABEs). Until now, a few groups have suggested improved versions of CBEs and ABEs by rendering a few mutations to cytidine deaminases (CDs) or adenosine deaminases (ADs). In this manuscript, the authors identified sites tolerant in nCas9 through the MuA-transposon-based genetic screen and constructed Cas Embedded (CE) versions with various CDs and ADs, which ultimately showed minimized off-target effects for both CBEs and ABEs. I think that this approach is novel and attractive because CE construct can be commonly used for any fusion proteins. Overall, the topic of this manuscript is potentially suitable for this journal, Nature Communications, but it should be largely revised to strengthen this study.

1) The key idea of this study is to insert the CD/AD enzymes into the middle of nCas9, resulting in reduced DNA or RNA off-target effects. But, any suggested reasons why the new constructs reduced the off-target effects are not described throughout the main text.

Indeed, the sgRNA-independent DNA or RNA off-target effects are thought to be caused by CD and AD enzymes, rather than nCas9 nuclease. Hence, the insertion of CD/AD enzymes into the middle of nCas9 might hinder the CD/AD activity partly or reduce the degree of freedom of the enzymes, so that the CD/AD can hardly work without the help of nCas9.

For example, what is the size of the linker between inserted CD/AD enzymes and nCas9? If the linker become larger and more flexible, CE-CBE/ABE variants might generate the sgRNA-independent DNA or RNA off-target effects. It should be deeply addressed.

2) The CE strategy seems to be very attractive because it can be commonly used for any fusion proteins including CD/AD enzymes. To emphasize the utility of it, further comparison experiments with other versions seem to be required. For example, in Figure 1, the comparison between CE-ABE variants and YE1-BE3 or BE3_R126E [Ref #8] is necessary in terms of specificity and activity.

3) Another study reported that ABEs can also catalyze cytosine conversions in confined TCN sequence context [Kim et al., Nat Biotechnol. 37, 1145-1148, 2019], which has not been addressed, so far. I am curious whether the CE-ABE variants further affect the cytosine conversion activity.

4) The discussion part seems to be poor in this manuscript. Researchers have tried to engineer Cas9. For example, a circularly permuted Cas9 (Cas9-CP) [Oakes et al. Cell 176, 254-267, 2019] was utilized for base editing [Huang et al., Nat Biotechnol. 37, 626-631, 2019]. The introduction and comparative discussion can reinforce this study.

5) This version of manuscript seems to be a "Letter" format. It should be revised to be an "Article" format. In the meantime, it might be better to increase main figures from two to three or four by adding most data in Supplementary Figure 2~4 that are valuable to be shown in main text. In addition, Figure 2b should be mentioned in main text prior to Figure 2c and 2d, or the configuration of Figure 2 should be re-arranged.

6) In Supplementary Figure 5a, the figure legend indicates the usage of Cas-OFFinder and CRISPROR, but corresponding figure indicates Cas-OT and Cas-OFFinder. In addition, the searching condition of both Cas-OFFinder and CRISPROR should be denoted. It seems that CRISPROR predict much more off-target sites than Cas-OFFinder, but I guess the searching condition might be different between each other.

7) (minor) Some typos should be corrected.

- line 94, "CBES" -> "CBEs"
- line 112 and line 122, CE^{1048-1063-A3A(Y130)} -> CE^{1048-1063-A3A(Y130F)}
- line 114, negligble -> negligible
- line 129, mutagesis -> mutagenesis
- line 141, "optimizating" -> "optimizing", "primer editing" -> "prime editing"
- line 329, GPF -> GFP
- In lines 135~136, reference regarding on the phrase "inserting deaminase on the surface of dCasRx" is necessary.

Rebuttal

We are extremely grateful to the reviewers for scrutinizing the paper and for the valuable comments, which has led to marked improvement in paper quality. In particular, we have reformatted the paper from the letter into an article, which entails large reorganization. Thus, relevant supplementary data have been incorporated into the main figures, and the main text, especially the discussion section, much substantiated. The specific concerns are addressed below, where the sentences copied from the revised manuscript are highlighted in red.

Reviewer #1 (Remarks to the Author):

The authors of this manuscript have tested the interesting idea that Cas9-internal insertion of deaminase domains (rather than N- or C-terminal fusions) can produce base editors with superior properties. Neutral insertion sites were identified by an elegant transposon-based genetic screen. Testing both cytidine and adenosine deaminases, the authors were indeed able to create base editors with reduced off-target effects with respect to both RNA editing and DNA editing. Overall, I find the approach developed by the authors very interesting. The new base editors certainly add useful tools to the genome editing toolbox, and, in the future, the approach likely can be extended to other Cas fusion proteins.

Thank a lot for your enthusiasm and interest!

Major points:

1. The authors tested their approach as a strategy to potentially reduce the off-target effects of BEs. However, all generated CE-ABEs do not outperform the previously reported ABE7.10F148A in terms of reducing RNA off-targets or improving on-target editing efficiency. Considering that ABE7.10F148A and the new CE-ABEs also have different base editing windows, it seems desirable to test a few more target sites and compare editing efficiency and precision.

Thanks for the advice, but we are hesitant to do the suggested experiment, because ABE7.10^{F148A} is not a focus of our paper. Yes, ABE7.10^{F148A} is highly precise, creating fewer RNA off-target edits than CE¹⁰⁴⁸⁻¹⁰⁶³-ABE (the optimal CE-ABE), and its editing window is also presumably narrower. However, ABE7.10^{F148A}, just like ABE7.10, is far less efficient than ABEmax, an improved version of ABE7.10 (Gaudelli et al., 2017; Zhou et al., 2019)(Koblan et al., 2018). Importantly, CE¹⁰⁴⁸⁻¹⁰⁶³-ABE (the optimal CE-ABE) is as efficient as ABEmax (Fig. 2). We can thus infer that ABE7.10^{F148A} is weaker than CE¹⁰⁴⁸⁻¹⁰⁶³-ABE, which we have actually directly observed at a site tested (Fig. 2d). Thus, ABE7.10^{F148A} is more specific but less active than CE¹⁰⁴⁸⁻¹⁰⁶³-ABE. We feel that comparing the two editors at a few more sites are unlikely to alter this conclusion.

As alluded to above, ABEmax is an updated version of ABE7.10, with a much higher editing efficiency. It would be interesting to see whether introducing F148A into ABE4max can similarly minimize its massive RNA off-target editing, but we hope you would agree that this is beyond the scope of the current study, particularly considering the rapid pace of the field. Furthermore, in contrast to ABEs, we show that CE¹⁰⁴⁸⁻¹⁰⁶³-A3A(Y130F) has clear advantages

over the existing CBEs, thus proving the point that CE can create useful reagents for gene editing.

We therefore wish that you would agree that testing the additional sites for CE-ABE is not essential.

2. Figure 1a: How many colonies were obtained by the double selection and how many of them were sequenced? This is important to know to be able to judge whether or not the screen has been saturating. Also, Oakes et al. (2016) employed a similar screening strategy and also identified tolerant insertion sites. The combined information on tolerant sites should be discussed with regard to possible saturation and future engineering options.

We have performed a total of three screens, obtaining 45, 40, 37 colonies, respectively. Some insertion sites were recovered multiple times in almost every screen, suggesting that the screening has been saturating. Interestingly, the insertional sites have little overlap with the 175 hotspots described by Oakes et al, presumably reflecting the differences in the experimental design, as they search for dCas9 sites tolerant of PDZ insertion, and their screen readout is dCas9-mediated gene silencing. The two studies collectively suggest that although Cas9 tolerates insertions at numerous sites, the optimal sites may be protein-specific. Nevertheless, the region between aa 1048-1063 identified in our study may be generally tolerant of diverse proteins, because this site is non-conserved and supportive of superior performance of both ABE and CBE. This region could therefore serve as the starting point when using the CE strategy in the future.

Per your advice, we have incorporated the above information into the Discussion section (Line 243-261).

3. In Figure 2, the authors compare the editing efficiencies of CE1048-1068-A3A(Y130F) and YE1-BE4max in a C-rich sequence. Considering that the YE1 variant has recently been reported to reduce DNA off-targets, it is unfortunate that the authors have not included YE1 as a comparator for DNA off-targets.

Thanks for the advice, but we have refrained from the experiment for the following reasons:

1) Off-target editing by CE¹⁰⁴⁹⁻¹⁰⁶³-A3A(Y130F) was minimal, as GOTI detected only 43 SNVs. Although YE1-BE4max might prove even more specific, the putative advantage would be eclipsed by the much higher efficiency of CE¹⁰⁴⁹⁻¹⁰⁶³-A3A(Y130F) at methylated and C-rich targets. Thus, for editing such targets, CE¹⁰⁴⁹⁻¹⁰⁶³-A3A(Y130F) is clearly preferable over YE1-BE4max, regardless of the off-target effects of YE1-BE4max.

2) We agree that it would be ideal to do GOTI on YE1-BE4max. Unfortunately, GOTI is very labor-intensive and costly, and in this paper, we have already performed GOTI for as many as 6 editors, making additional assays real challenging.

3) We have previously performed GOTI assay for YE1-BE3, showing that its off-target

editing is undetectable (Zuo et al., 2019). We have now added this data as a proxy for YE1-BE4max (new Fig. 4a-b). Of note, YE1-BE4max would presumably create more SNVs than BE3-YE1, given its much higher efficiency.

Minor points:

1. In Figure 1a, the authors write “LB with Kan 37 □ 5h”. In the text, they write “cells grown overnight on plates” (l. 43) and in the Methods section it is 16 hours (l. 199). These inconsistencies should be removed.

Thanks a lot for the scrutinization, and our apologies for the oversight. The procedure has been clarified in part by optimizing the workflow illustration in Fig. 1a. Briefly, after electroporation, the cells were grown overnight Kan+ plate. The colonies were then scraped and pooled, and subjected to deep-seq to define the insertion sites in unscreened library. The pooled colonies were also transferred to liquid media containing IPTG for nCas9 induction and correction of the premature stop codon at AmpR gene. The cells were then plated on Amp+Kan+ plates and colonies individually sequenced to map the insertion sites in the recovered plasmids.

2. Line 96: The GOTI method was developed by Zuo et al., (Science, 2019), which should be cited here.

Thanks! We have added the reference per your advice.

3. There are many other instances, where it would be appropriate to cite additional references. For example, in the last sentence of the text, prime editors are mentioned without a reference (Anzalone et al., Nature 2019). Also, previous work on the design of base editors with improved properties should be discussed in the Introduction and/or Discussion section of the manuscript (e.g., Kim, Y. B., et al. Increasing the genome-targeting scope and precision of base editing with engineered Cas9-cytidine deaminase fusions. Nat. Biotechnol. 35, 371-376, 2017; Tan, J., et al. Engineering of high-precision base editors for site-specific single nucleotide replacement. Nat. Commun. 10, 439, 2019; Tan, J., et al. Expanding the genome-targeting scope and the site selectivity of high-precision base editors. Nat. Commun. 11, 629, 2020). Overall, the writing is quite compact, and clarity and intelligibility would benefit from a few extra sentences here and there.

Thanks for the advice! The paper was actually submitted in a rush. We have now included all the references you cited, and beyond.

4. Figure 1e: lettering is too small to be legible (especially the letters and numbers in subscript and superscript). Also, in the legend, it is unclear what the “n” refers to.

We have enlarged the font size as advised. In the legend, sentence mentioning “n” is an error and now deleted. Sorry for the oversight.

5. Fig. S1d: ditto—font size needs to be increased.

We have enlarged the font size as advised.

6. Fig. S6: What does Tyr mean?

Sorry for the lack of clarity. Tyr is the mouse gene encoding tyrosinase, a copper-containing oxidase that functions in the formation of pigments. We have given the full gene name in the revision.

7. Line 100, “Supplementary Fig 6a” should be “Supplementary Fig 6”.

Thanks! The figure has been incorporated into Fig. 4 in the revision.

8. Line 122, “A3A(Y130)” should be “A3A(Y130F)”.

We have made the correction. Thanks!

9. Line 129, “mutagesis” should be “mutagenesis”, l. 15, “an obstacle”, l. 257, “zona pellucida” (not: zonae pellucidae), l. 141: “prime editor” (not: primer editor), etc. – there are numerous other small language-related problems, and overall, the manuscript could do with a final read by a native English speaker.

Sorry for the language issues that mostly occurred due to the rush in manuscript preparation.

We have tried to correct them all. Thanks!

10. Line 130: To not confuse the reader, it would be better to refer to point mutations “in the deaminase domain”. Also, deaminase truncations should be mentioned here (Tan, J. et al., 2019 and 2020).

We have made the correction and added the references as advised. These references were also added in the Introduction.

11. L. 98: “sgRNA targeting Tyr” – do the authors mean “a codon for Tyr”?

Sorry for the lack of clarity. Tyr is the mouse gene encoding tyrosinase, a copper-containing oxidase that functions in the formation of pigments. We have given the full gene name in the revision.

12. L.75f: I don’t think a 10% reduction in on-target editing can be called “uncompromised”

The reduction is actually 3% (from 92% to 89%). Nevertheless, we have replaced “uncompromised” with “comparable” to avoid confusion.

13. L.106: A 30-40% reduction in off-target mutations should not be referred to as “dramatic”

Sorry for the confusion.

1) The percentages described here are not the mutation rates of the editors, but the frequencies of C>G>T:A (G>A and C>T) SNV subsets among the total SNVs observed in the embryos. These SNV subsets result from cytidine deamination, and thus reflect the off-target effects of endogenous Apobec enzymes in the case of GFP treated embryos and the endogenous enzymes in conjunction with the exogenous base editors in other embryos. Since the frequency for AncBE4max, CE-CBEs and GFP is 92%, 57-68% and 43%, respectively, the

SNVs induced by AncBE4max and CE-CBEs constitute 49% and 14-25%, respectively, representing 2x-3.5x reduction in the CBE induced SNVs.

2) Paradoxically, the total SNVs for embryos treated with AncBE4max, CE-CBEs and GFP are 773, 19-31 and 18, respectively, indicating far more dramatic decreases in off-target edits in CE-CBE relative to AncBE4max. The discrepancy apparently arises because the SNVs are so rare in CE-CBEs and GFP, making it impossible to accurately estimate the frequencies of G>A or C>T among the few SNVs detected. The SNV frequencies are therefore intended only for qualitative assessment of the off-target effects.

We have updated the text, where “dramatically” was replaced with “markedly” to avoid confusion (Line 171-181):

in the SNVs detected in AncBE4max-treated embryos, the SNV subsets resulting from C:G>T:A conversion predominated, constituting 92% of the total SNVs, whereas in the embryos treated with CE-CBEs, GFP or BE3-YE1, the subsets were less abundant (<68%, but accurate quantification not feasible due to the scarcity of SNVs; Fig. 4a, bottom and Supplementary Fig. 2c). The same trend was observed for A3A-editors, with substantial numbers of SNVs (276) detected in BE3-A3A(Y130F) but much less (43-63) in CE¹⁰⁴⁸⁻¹⁰⁶³-A3A(Y130F) or CE¹⁰⁷²-A3A (Y130F) (Fig. 4a and Supplementary Figs. 2). We conclude that CE markedly decreased DNA off-target effects of CBEs.

Reviewer #2 (Remarks to the Author):

General comments;

DNA base editors, which include cytosine base editors (CBEs) and adenine base editors (ABEs), are promising tools and have been harnessed to many research areas such as therapeutic editing. However, several studies showed shortcomings of the tools, including promiscuous deamination effects in both DNA and RNA (CBEs) and in DNA (ABEs). Until now, a few groups have suggested improved versions of CBEs and ABEs by rendering a few mutations to cytidine deaminases (CDs) or adenosine deaminases (ADs). In this manuscript, the authors identified sites tolerant in nCas9 through the MuA-transposon-based genetic screen and constructed Cas Embedded (CE) versions with various CDs and ADs, which ultimately showed minimized off-target effects for both CBEs and ABEs. I think that this approach is novel and attractive because CE construct can be commonly used for any fusion proteins. Overall, the topic of this manuscript is potentially suitable for this journal, Nature Communications, but it should be largely revised to strengthen this study.

Thanks a lot for your appreciation of the CE strategy! We have revised the MS per your advice.

1) The key idea of this study is to insert the CD/AD enzymes into the middle of nCas9, resulting in reduced DNA or RNA off-target effects. But, any suggested reasons why the new constructs reduced the off-target effects are not described throughout the main text.

Indeed, the sgRNA-independent DNA or RNA off-target effects are thought to be caused by CD and AD enzymes, rather than nCas9 nuclease. Hence, the insertion of CD/AD enzymes

into the middle of nCas9 might hinder the CD/AD activity partly or reduce the degree of freedom of the enzymes, so that the CD/AD can hardly work without the help of nCas9. For example, what is the size of the linker between inserted CD/AD enzymes and nCas9? If the linker become larger and more flexible, CE-CBE/ABE variants might generate the sgRNA-independent DNA or RNA off-target effects. It should be deeply addressed.

Thanks a lot for the insightful comments! We have added the following in the Discussion section based on your advice (Line 218-230):

How exactly CE reduces the off-target effects is unclear. Off-target editing of DNA base editors are caused by the deaminase moiety independent of nCas9, whereas the on-target editing is achieved when the deaminase moiety is brought into the proximity of the targets by nCas9. Embedding the deaminase domains in nCas9 might create some steric hindrance that hampers their free access to the off-targets, but not an on-target thanks to their forced tethering via nCas9. This idea is testable by replacing the current XTEN linkers in the CE-editors with longer, flexible linkers, which could relieve the putative steric hindrance. Besides, structural studies as exemplified by Doudna and colleagues²¹ should be valuable for providing conclusive evidence regarding the mechanisms of CE. Of note, circularly permuted nCas9 have been created whose protein termini are placed at various positions²². It would be interesting to determine whether fusing the deaminase domains to the termini could also selectively compromise off-target editing.

2) The CE strategy seems to be very attractive because it can be commonly used for any fusion proteins including CD/AD enzymes. To emphasize the utility of it, further comparison experiments with other versions seem to be required. For example, in Figure 1, the comparison between CE-ABE variants and YE1-BE3 or BE3_R126E [Ref #8] is necessary in terms of specificity and activity.

Thanks for your appreciation of the CE strategy! It is indeed important to extensively compare the CE-editors with the existing ones. Therefore, we have compared CE-ABE with ABEmax (Fig.2), CE-CBE with AncBE4max (Fig. 3a-c), and CE-A3A(Y130F) with BE4max-A3A(Y130F) (Fig. 3d-h). ABEmax and AncBE4max were picked for comparison because they represent the most active ABE and CBE at the time, whereas BE4max-A3A(Y130F) was specifically generated in the current study for the comparison with CE-A3A(Y130F). Besides, both the efficiency and specificity of these editors were compared, the former at a total of 31 randomly selected target sites in human HEK 293T and mouse N2a cell lines, and the latter in HEK293T cells (for RNA off-targets) and mouse embryos (for DNA off-targets, via GOTI). These data collectively show CE is indeed an attractive complement to the conventional mutagenesis approach for base editor optimization.

Regarding Fig. 1, it is focused on CE-ABE and so we compared it with ABEmax, but not a CBE such as YE1-BE3. To avoid confusion, all the data on CE-ABE has now been presented in Fig. 2, and Fig. 1 now describes only the genetic screen, while Fig. 3-4 exclusively CBE. Finally, we have previously performed GOTI on YE1-BE3 (Zou et al, 2019), and have now added the data to Fig. 4a and b in response to Reviewer 1's comment.

3) Another study reported that ABEs can also catalyze cytosine conversions in confined TCN sequence context [Kim et al., Nat Biotechnol. 37, 1145-1148, 2019], which has not been addressed, so far. I am curious whether the CE-ABE variants further affect the cytosine conversion activity.

Thanks for your curiosity. We have revisited the editing data. Among the 21 sites tested, only three carry the TCN motif within the editing window, with a total of 4 targeted Cs in this motif. A minority of Cs were indeed deaminated at each site by both CE¹⁰⁴⁸⁻¹⁰⁶³-ABE and ABEmax (figure below). Interestingly, the reaction products were somehow predominantly T for CE¹⁰⁴⁸⁻¹⁰⁶³-ABE, but a mixture of T, G and A for ABEmax. We prefer not to show this data because it is too preliminary; a lot more sites are perhaps needed to make a solid conclusion.

4) The discussion part seems to be poor in this manuscript. Researchers have tried to engineer Cas9. For example, a circularly permuted Cas9 (Cas9-CP) [Oakes et al. Cell 176, 254–267, 2019] was utilized for base editing [Huang et al., Nat Biotechnol. 37, 626-631, 2019]. The introduction and comparative discussion can reinforce this study.

Thanks for the comments. Per your advice, we have substantiated the Discussion section, partly by mentioning the Cas9-CP data (Line 230-233):

circularly permuted nCas9 have been created whose protein termini are placed at various positions(Oakes et al., 2019). It would be interesting to determine whether fusing the deaminase domains to the termini could also compromise off-target editing

5) This version of manuscript seems to be a “Letter” format. It should be revised to be an “Article” format. In the meantime, it might be better to increase main figures from two to three or four by adding most data in Supplementary Figure 2~4 that are valuable to be shown in main text. In addition, Figure 2b should be mentioned in main text prior to Figure 2c and 2d, or the configuration of Figure 2 should be re-arranged.

We have reformatted the letter into an Article, and reconfigured the figures essentially per your advice.

6) In Supplementary Figure 5a, the figure legend indicates the usage of Cas-OFFinder and CRISPROR, but corresponding figure indicates Cas-OT and Cas-OFFinder. In addition, the searching condition of both Cas-OFFinder and CRISPROR should be denoted. It seems that

CRISPROR predict much more off-target sites than Cas-OFFinder, but I guess the searching condition might be different between each other.

Sorry for the oversight. We used Cas-OT and Cas-OFFinder to identify potential gRNA-dependent off-target sites for gRNAs with the NRG PAM. The searching parameter for Cas-OT was set as 2 and 3 mismatches at the seed (the first 12nt adjacent to PMA) and non-seed (the 8 distal sequences) regions, respectively. For Cas-OFFinder, which does not discriminate between seed vs. non-seed regions, it was set as 3 or fewer mismatches along the 20-nt protospacer. We have updated the figure legends and Methods accordingly. Thanks!

7) (minor) Some typos should be corrected.

- line 94, “CBES” -> “CBEs”

We have corrected it. Thanks!

- line 112 and line 122, CE^{1048-1063-A3A(Y130)} -> CE^{1048-1063-A3A(Y130F)}

We have corrected it. Thanks!

- line 114, negligble -> negligible

We have corrected it. Thanks!

- line 129, mutagesis -> mutagenesis

We have corrected it. Thanks!

- line 141, “optimizating” -> “optimizing”, “primer editing” -> “prime editing”

We have corrected it. Thanks!

- line 329, GPF -> GFP

We have corrected it. Thanks!

- In lines 135~136, reference regarding on the phrase “inserting deaminase on the surface of dCasRx” is necessary.

We have corrected it. Thanks!

References

Gaudelli, N.M., Komor, A.C., Rees, H.A., Packer, M.S., Badran, A.H., Bryson, D.I., and Liu, D.R. (2017). Programmable base editing of A•T to G•C in genomic DNA without DNA cleavage. *Nature* 551, 464–471.

Koblan, L.W., Doman, J.L., Wilson, C., Levy, J.M., Tay, T., Newby, G.A., Maiani, J.P., Raguram, A., and Liu, D.R. (2018). Improving cytidine and adenine base editors by expression optimization and ancestral reconstruction. *Nature Biotechnology*.

Zhou, C., Sun, Y., Yan, R., Liu, Y., Zuo, E., Gu, C., Han, L., Wei, Y., Hu, X., Zeng, R., et al. (2019). Off-target RNA mutation induced by DNA base editing and its elimination by mutagenesis. *Nature* 571, 275–278.

Zuo, E., Sun, Y., Wei, W., Yuan, T., Ying, W., Sun, H., Yuan, L., Steinmetz, L.M., Li, Y., and Yang, H. (2019). Cytosine base editor generates substantial off-target single-nucleotide variants in mouse embryos. *Science* 364, 289–292.

Reviewers' Comments:

Reviewer #1:

Remarks to the Author:

The manuscript has much improved, and is now also easier to read. There are still a number of typos and grammatical errors that should be fixed, e.g.:

- l. 227: concusive
- l. 238: editing at genomic DNA; better: editing on genomic DNA
- l. 248: total 3 screens; rather: a total of 3 screens (or: in total 3 screens)
- l. 476: editing rates of the susceptible as at individual sites; ??? "susceptible As at individual sites"?

Reviewer #2:

Remarks to the Author:

The authors have mostly answered the issues I raised in the earlier review. However, I would note that the description on ABEmax or CE-ABE mediated cytosine editing activities may be informative in terms of high-fidelity ABE construction. I agree that the data shown in Response #3 is too preliminary but, it is notable that both ABEmax and CE-ABE as a current version generated cytosine editing. It would be better to describe it at least in the discussion part.

Rebuttal

Reviewer #1 (Remarks to the Author):

The manuscript has much improved, and is now also easier to read. There are still a number of typos and grammatical errors that should be fixed, e.g.:

- l. 227: concusive

Fixed. Thanks!

- l. 238: editing at genomic DNA; better: editing on genomic DNA

Corrected as advised. Thanks!

- l. 248: total 3 screens; rather: a total of 3 screens (or: in total 3 screens)

Corrected as advised. Thanks!

- l. 476: editing rates of the susceptible as at individual sites; ??? “susceptible As at individual sites”?

You meant L. 481. We have capitalized the A. Thanks!

Reviewer #2 (Remarks to the Author):

The authors have mostly answered the issues I raised in the earlier review. However, I would note that the description on ABEmax or CE-ABE mediated cytosine editing activities may be informative in terms of high-fidelity ABE construction. I agree that the data shown in Response #3 is too preliminary but, it is notable that both ABEmax and CE-ABE as a current version generated cytosine editing. It would be better to describe it at least in the discussion part.

Thanks for understanding. We have added the following in the discussion:

Of note, ABEs are known to catalyze cytosine conversions in confined TCN sequence context at very low frequencies²³, and the same seemed true for CE-ABE (not shown).